# An Empirical Evaluation of Federated Contextual Bandit Algorithms

## Abstract

As the adoption of federated learning increases for learning from sensitive data local to user devices, it is natural to ask if the learning can be done using implicit signals generated as users interact with the applications of interest, rather than requiring access to explicit labels which can be difficult to acquire in many tasks. We approach such problems with the framework of federated contextual bandits, and develop variants of prominent contextual bandit algorithms from the centralized seting for the federated setting. We carefully evaluate these algorithms in a range of scenarios simulated using publicly available datasets. Our simulations model typical setups encountered in the real-world, such as various misalignments between an initial pre-trained model and the subsequent user interactions due to non-stationarity in the data and/or heterogeneity across clients. Our experiments reveal the surprising effectiveness of the simple and commonly used softmax heuristic in balancing the well-know exploration-exploitation tradeoff across the breadth of our settings.

## 1 Introduction

Federated learning (Konečnỳ et al., 2016; McMahan et al., 2017; Kairouz et al., 2021b) has emerged as an important machine learning paradigm for settings where the raw training data remains decentralized across a potentially heterogeneous collection of devices. A key motivation for cross-device federated learning (henceforth FL) arises from scenarios where these devices belong to various users of a service, and the goal is to learn predictive models from the data generated when the user interacts with the service. This has benefits from a privacy perspective, and can also allow the development of more expressive models that leverage contextual features that would be unavailable in the datacenter.

Federated learning directly encodes the data minimization privacy principle including focused collection and ephemeral updates with immediate aggregation (Bonawitz et al., 2022), and much recent work has shown it is possible to combine these benefits with data anonymization for the trained models via differential privacy (McMahan et al., 2018; Kairouz et al., 2021a). Following standard terminology in this area, we refer to the end devices as the clients, and the learning process is orchestrated by a centralized server.

From a machine learning perspective, the bulk of federated learning algorithms can be effectively seen as a particular category of decentralized learning algorithms which aim to solve a loss minimization problem over the entire dataset distributed across the clients, without explicitly performing such a data collection (Wang et al., 2021). This is an attractive framework, as it captures many supervised machine learning settings, such as classification and regression, as long as the label signal arises naturally from user interaction. This is the case for example with next-word-prediction (Hard et al., 2018), where if the system does not correctly predict the next word, it is reasonable to assume the user will simply tap/swipe the desired word without using the next-word suggestions.

However, many real-world federated learning scenarios do not provide such a complete feedback signal. For example, consider an application where we want to display a featured image from a user's phone every time

they open the photo gallery. Other applications could be to annotate each image and/or text message with a label corresponding to its category from a predefined set, or to suggest emoji and stickers (where the user does not know the full set of options) in a mobile keyboard. In all these examples, the underlying training data for learning is highly sensitive to the user, and collecting ground truth labels from third-party human labelers is not feasible. Furthermore, even if privacy allowed the use of human labelers, in the first example of selecting a featured image, it is nearly impossible for a labeler to guess which image from a user's collection appeals to them, and it is impractical for a user to respond with the best choice of a featured image from their entire collection. A much more natural feedback modality in all these settings is to make a recommendation (of an image, label, emoji, or sticker) to the user, and observe and learn from their response to that recommendation. Further, note that both user preferences and the set of available recommendations may evolve over time. Supervised learning fails to properly capture such settings where we only observe feedback on the choices driven by the learning algorithm, and reinforcement learning (RL) offers a much better fit for these problems where we seek to learn from user feedback.

A particular subset of RL which is quite effective at capturing several recommendation settings is that of contextual bandits (CB) (Langford & Zhang, 2007; Auer, 2002; Agarwal et al., 2014). A key difference between RL/CB and more traditional supervised learning approaches is the explicit recognition that the algorithm only collects feedback for the choices it presents to the user, and hence it is important to navigate the *exploration/exploitation* tradeoff. That is, the algorithm should explore over a diverse set of plausibly good choices in any situation, and use the feedback to further prune the set of plausible choices. Motivated by the twin concerns of learning from user feedback in a decentralized and private manner, there is an emerging literature on federated CB learning (Huang et al., 2021; Dai et al., 2022; Dubey & Pentland, 2020). However, the bulk of the existing work is theoretical in nature, with a focus on simple models such as multi-armed or linear bandits, with a key focus on exploration in the federated setting. An important aspect of several works here is also developing the right notions of privacy suited to the interactive learning setting (Shariff & Sheffet, 2018; Dubey & Pentland, 2020).

In this work, we study federated CB learning with a complementary focus to the aforementioned works. We design federated adaptations of practical state-of-the-art CB algorithms from the centralized setting, and conduct an extensive empirical evaluation in a range of realistic settings. Algorithmically, we focus on a black-box approach, where we isolate a component of the centralized CB algorithms which relies on solving a classification or regression problem, and replace this with a federated learning counterpart. This is practically desirable, as it makes it easy to incorporate latest advances from federated optimization into the CB algorithms as drop-in replacements. The isolated federated optimization can also be combined with complementary privacy techniques such as secure aggregation (Bonawitz et al., 2017) and differential privacy (McMahan et al., 2018; Kairouz et al., 2021a).

We primarily consider cross-device FL (Kairouz et al., 2021b; Wang et al., 2021), which is typically more challenging than cross-silo FL due to constraints such as limited computation resources, bandwidth, and client availability; hence, our proposed framework and algorithms are also applicable to cross-silo FL, though more sophisticated cross-silo algorithms can be possibly designed if we relax the on-device constraints and tackle other challenges like feature alignment (Yang et al., 2019)

Even in the centralized setting, empirical evaluation of CB methods is limited to just a few works (Bietti et al., 2021; Foster et al., 2020), and often ignores practical concerns such as data non-stationarity and the impracticality of updating the CB model after each example. The federated setting adds further challenges related to data heterogeneity across clients, greater delays in model updates on clients and configuring the settings of the underlying federated optimization approach as some examples. Our work uses two popular FL benchmarks, `EMNIST` and `StackOverflow` (`SO` for short), and turns them into simulators for the federated CB setting by adapting and extending the ideas from the work of Bietti et al. (2021). Within this simulation, we evaluate federated adaptations of several centralized CB algorithms in both stationary and realistic simulations of non-stationary settings. We also study the influence of providing a small amount of labeled data to create an initial model, which is typical in practice.

Bietti et al. (2021) observed that the greedy approach offers an extremely strong baseline in stationary centralized settings. We show this result can extend to the federated setting, and in particular that a greedy strategy is highly effective when the problem is stationary and the model can be updated frequently. However, exploration becomes critical under delayed updates and/or non-stationarity. The use of a strong initial model can mitigate this to a reasonable degree, particularly in stationary settings. When exploration strategies are necessary, we find federated versions of a simple softmax exploration strategy, and an adaptation of FALCON, to be the best performing across the range of settings, with softmax being easier to tune than FALCON.

We emphasize our goal is not to show that bandit algorithms "win" against baselines. Rather, we hope that this study can both provide a valuable resource in terms of a strong evaluation setup for future research on federated CBs, as well as offer practical recipes for practitioners facing the federated CB setting and needing to decide whether the additional complexity of deploying a bandit algorithm with an explicit exploration strategy is likely to be beneficial.

## 2 Preliminaries

We begin by briefly recalling the federated learning and contextual bandit paradigms in this section. We then build on these to set up the federated contextual bandit setting in the next section.

### 2.1 Federated Learning

In a federated learning problem, we are given a distribution $p$ over a population $\mathcal{C}$ of clients. Client $c \in \mathcal{C}$ has an associated data distribution $D_c$ over samples $z \in \mathcal{Z}$. The learning algorithm aims to find a good model $f \in \mathcal{F}$ under some loss function $\ell : \mathcal{F} \times \mathcal{Z} \to \mathbb{R}$, so as to minimize the objective:

$$\min_{f \in \mathcal{F}} \mathbb{E}_{c \sim p} \mathbb{E}_{z \sim D_c} \ell(f, z). \tag{1}$$

Like most learning algorithms, the objective (1) is optimized approximately using a sample-based approximation. Unique to federated learning, however, the datasets stay local to each client, while model updates from each client are aggregated and applied on the central server. For intuition, a canonical federated learning algorithm is FEDAVG (McMahan et al., 2017), in which a random subset of the clients each use their local data to compute an update to the shared model by performing a few stochastic gradient steps with the local dataset. The updates are then communicated to the server which averages these local model changes and uses this average to update the shared global model at the server.

### 2.2 Contextual Bandits

Contextual bandits are a paradigm to learn from interaction data where each interaction consists of observing a context $x \in \mathbb{R}^d$ from some fixed and unknown distribution, choosing an action $a \in \mathcal{A}$ from some action set $\mathcal{A}$ and observing some reward $r(x, a) \in \mathbb{R}$ specifying the quality of the action $a$ for context $x$. Crucially, the learner receives no signal on the quality of actions $a' \neq a$ which were not chosen. We let $D$ represent the joint distribution of $(x, r)$, but also overload it to denote the marginal distribution over $x$, when it is clear from the context. We view $r$ as a random variable, where $r(x, a)$ is the $a_{th}$ entry in the reward vector $r \sim D(\cdot|x)$. As mentioned above, the learner never observes the full reward vector $r$, but only the entry $r(x, a)$ when it chooses action $a$ upon observing context $x$. The learner has access to a policy class $\Pi \subseteq \{\mathcal{X} \to \mathcal{A}\}$, where a policy is a mapping from contexts to actions. For deterministic policies we write $\pi(x) \in \mathcal{A}$; we generalize this to randomized policies where $\pi(a|x) \in [0, 1]$ below. The goal of learning is to find a policy that maximizes the expected reward, and the quality of some policy $\pi$ is measured in terms of regret, defined as[1]

$$\text{Regret}(\pi) = \mathbb{E}_{(x,r) \sim D}[r(x, \pi(x))] - \max_{\pi' \in \Pi} \mathbb{E}_{(x,r) \sim D}[r(x, \pi'(x))]. \tag{2}$$

---

[1]For a randomized policy, we can replace $\pi(x)$ with an expectation over $a \sim \pi(\cdot|x)$.

For intuition, a deterministic policy class $\Pi$ might be induced by a regression function class $\mathcal{F}$ as $\Pi = \{\pi_f \; : \; \pi_f(x) = \text{argmax}_{a \in \mathcal{A}} f(x, a), f \in \mathcal{F}\}$, where the functions $f$ are trained to predict the expected reward using regression. That is, given a dataset $(x_s, a_s, r_s)_{s=1}^{t-1}$ of historical examples (where $r_s \in \mathbb{R}$ represent the realization of the random variable $r(x_s, a_s)$), we train the reward estimator $f_t = \text{argmin}_{f \in \mathcal{F}} \sum_{s=1}^{t-1} (f(x_s, a_s) - r_s)^2$. A common choice we will use in most of our setup is when the functions $f$ are parameterized as $f_\theta$ for some parameter $\theta \in \Theta$, where $\theta$ might denote the weights of a linear function or a neural network, for instance.

There are several standard ways of extracting a randomized policy $\pi_t$ from $f_t$, some of which we discuss below.

- `Greedy` corresponds to the standard supervised learning approach, where we always choose the best action according to $f_t$,

$$\Pi = \left\{ \pi_f \; : \; \pi_f(a|x) = 1 \text{ if } a = \underset{a' \in \mathcal{A}}{\text{argmax}} \, f(x, a') \text{ and } 0 \text{ otherwise}, f \in \mathcal{F} \right\} \tag{3}$$

  (with ties broken arbitrarily).

- $\epsilon$-`Greedy` chooses the greedy action with probability $1 - \epsilon$, and with probability $\epsilon$, picks an action uniformly from $\mathcal{A}$. The extra exploration helps in collecting a more diverse dataset to train $f_t$, with the parameter $\epsilon$ providing a tradeoff between exploration and exploitation. For any data distribution $D$, the regret of $\epsilon$-`Greedy` is known to be bounded, whenever the class $\mathcal{F}$ is sufficiently expressive (Langford & Zhang, 2007; Agarwal et al., 2012).

- `Softmax` is another variant of `Greedy`, where the policy uses a softmax distribution on the predicted rewards by the underlying model: $\pi_t(a|x) \propto \exp(f(x, a)/\beta)$. When $\beta$ approaches zero, the $\pi_t$ approaches the greedy policy, and diffuses to a uniform exploration for $\beta = \infty$. In general, this strategy does not have theoretical guarantees on the regret, but is often practically used owing to its simplicity.

- `FALCON` is provably optimal in the worst-case (Simchi-Levi & Xu, 2022; Foster & Rakhlin, 2020) and uses a more carefully crafted distribution over actions, given $f_t$ (see line 6 in Algorithm 2). The degree of exploration is governed by two hyperparameters $\gamma$ and $\mu$, which makes this strategy a little harder to tune in practice. For setting these hyperparameters, we depart from the theoretical recommendation in Simchi-Levi & Xu (2022) of using a careful schedule and use a best constant setting closer to Foster & Rakhlin (2020), as some of the quantities in the theoretical recommendations depending on the function class complexity and failure probability are unknown in practice.

## 3 Federated Contextual Bandits

We begin with the high-level problem setting and the algorithmic framework. We then present detailed federated variants of popular CB algorithms.

### 3.1 Problem Setting

With both the FL and CB settings defined individually, we now describe the federated CB setting. The high-level framework for the algorithms and the interaction with the environment is presented in Algorithm 1. In a federated CB problem, there is a distribution $p$ over clients $c \in \mathcal{C}$, with each client having a joint distribution $D_c$ over context and reward pairs. The server maintains a global policy $\pi \in \Pi$, which is now learned in a federated manner. That is, each client maintains some (potentially stale) version of the server's policy locally, which we denote as $\pi_c$.[2] Each client $c$ collects data by choosing actions on observed contexts

---

[2]Federated learning makes the possibility of learning a personalized policy $\pi_c$ much easier, but we focus on global policies in this work. Note that even with a global policy, the contextual features $x$ can be used to encode substantial information about the particular client $c$, and hence can lead to strongly personalized recommendations.

---
**Algorithm 1** Federated Contextual Bandits
---
**Require:** Communication rounds $T$ per period; training periods $I \geq 1$; initial inference model $\theta_0$

 1: **for** $i = 1, 2, \ldots, I$ **do**

 2:     Deploy inference policy $\pi$ parameterized by $\theta_{i-1}$ to all clients $\mathcal{C}$

 3:     **for** each $c \in \mathcal{C}$ **in parallel do**

 4:         $B_c \leftarrow \text{BanditInference}(\pi, \theta_{i-1})$                                       $\triangleright$ Algorithm 2

 5:     **end for**

 6:     $\triangleright$ In a real deployment, training and inference might occur in parallel, but we simulate sequentially:

 7:     Initialize optimization $\theta^{(0)} \leftarrow \theta_{i-1}$

 8:     **for** $t = 1, 2, \ldots, T$ **do**

 9:         $\theta^{(t)} \leftarrow \text{FederatedRound}(\theta^{(t-1)})$                                $\triangleright$ Algorithm 3

10:     **end for**

11:     $\theta_i \leftarrow \theta^{(T)}$

12: **end for**
---

according to $\pi_c$ and logs the reward received (lines 3-5 in Algorithm 1), and we call this operation *bandit inference*. Some subset of the clients periodically participate in *federated training* to update the policy $\pi$ at the server, using their local data (lines 7-11). We explain the details of inference and training rounds in detail below.

**Bandit inference.** Bandit inference refers to the user-visible use of the policy $\pi_c$ locally at a client, whenever it is queried for an action with a context. For instance, this might correspond to choosing a featured image or an emoji recommendation upon observing the user's photo album or text message in our previous examples. Formally, at an inference step, a client $c$ observes a context $x \sim D_c$, chooses an action $a \sim \pi_c(\cdot|x)$ and observes the reward $r \sim D(\cdot|x, a)$. The inference steps happen asynchronously at the clients and do not require any communication, since the client only invokes a locally stored version of the policy to choose actions. The agent also maintains an internal log of inference tuples of the form $(x, a, r, \pi_c(a|x)) \in (\mathbb{R}^d, \mathcal{A}, \mathbb{R}, [0, 1])$, which are saved in data cache (Hard et al., 2018) and later used to update the server policy in the training rounds which we describe next.

**Federated training.** Periodically, the server polls a subset of the clients to participate in federated training. Roughly, this corresponds to using the inference logs across the participating clients to improve the regression model $f(\cdot, \cdot; \theta)$. However, this federated training for policy improvement happens in a decentralized manner with no explicit data pooling. For instance, each participating client $c$ downloads the current server regression parameters $\theta^{(t)}$ and uses its local logs to compute a local gradient direction, which is communicated to the server. The server then accumulates the gradients across the clients to update $\theta^{(t)}$ to form $\theta^{(t+1)}$. After several communication rounds, the training period concludes and the server can broadcast the updated regression parameters (and hence updated policy) to all the clients, or rely on the clients to pull an updated policy periodically.

**Federated regret.** The typical performance metric in a centralized CB setting is regret, which measures the average performance of a policy $\pi$ when we perform inference according to it, relative to the best possible policy in some class $\Pi$ as in Eq. (2). Analogously in the federated CB problem, we seek to control the bandit inference regret. Informally, federated regret is the average expected regret that is incurred across the client population, during the bandit inference process. To define the metric formally, let us consider a policy $\pi$ which is used simultaneously by a population of clients distributed according to $p$. Then the average federated inference regret of this policy $\pi$, relative to a policy class $\Pi$ is given by:

$$\text{Fed-Regret}(\pi) = \mathbb{E}_{c \sim p} \mathbb{E}_{(x,r) \sim D_c, a \sim \pi(\cdot|x)}[r(x, a)] - \max_{\pi' \in \Pi} \mathbb{E}_{c \sim p} \mathbb{E}_{(x,r) \sim D_c, a \sim \pi'(\cdot|x)}[r(x, a)]. \tag{4}$$

We evaluate a common policy $\pi$ at all the clients, instead of allowing a separate policy $\pi_c$ at each client, because we do not consider client-specific policies in this work. The client policies might further differ in practical settings, where the client might poll parameters from the server asynchronously, and we omit such generalizations here to focus on the key ideas in the presentation.

### 3.2 Federated CB algorithms

---

**Algorithm 2** Bandit Inference on Client $c$

---

**Require:** Model parameters $\theta$; number of actions $K = |\mathcal{A}|$; data cache size $M$; exploration parameter $\epsilon$ for $\epsilon$-Greedy , $\beta$ for Softmax , $\mu, \gamma$ for FALCON

1: (Optional) initialize data cache $B_c = \emptyset$      $\triangleright$ The cache can be reset for simplicity in simulation
2: **for** $j = 1, \ldots, M$ **do**      $\triangleright$ We only simulate sufficient user interactions to fill the cache
3:      Observe $x^j \sim D_c$. Let $a_\theta^j = \operatorname{argmax}_{a \in \mathcal{A}} f_\theta(x^j, a)$
4:      $\pi(a|x^j) = 1 - \epsilon + \epsilon/K$ if $a = a_\theta^j$ else $\epsilon/K$      $\triangleright$ $\epsilon$-Greedy
5:      $\pi(a|x^j) = \exp(f_\theta(x^j, a)/\beta)/\sum_b \exp(f_\theta(x^j, b)/\beta)$      $\triangleright$ Softmax
6:      $\pi(a|x^j) = 1/\left(\mu + \gamma(f_\theta(x^j, a_\theta^j) - f_\theta(x^j, a))\right)$ if $a \neq a_\theta^j$ else $1 - \sum_{b \neq a_\theta^j} \pi(b|x^j)$      $\triangleright$ FALCON
7:      Sample $a^j \sim \pi(\cdot|x^j)$ and observe $r^j$ for $a^j$
8:      $B_c \leftarrow B_c \cup \left\{(x^j, a^j, r^j, \pi(a^j|x^j))\right\}$
9: **end for**
10: **return** $B_c$

---

**Algorithm 3** One Round of Federated Optimization

---

**Require:** Global model $\theta^{(t-1)}$ from the previous round; subset of clients $\mathcal{S}^{(t)} \subset \mathcal{C}$
1: Broadcast $\theta^{(t-1)}$ from server to clients $\mathcal{S}^{(t)}$
2: **for** each $c \in \mathcal{S}^{(t)}$ **in parallel do**
3:      $\Delta_c^{(t)} = \text{ClientUpdate}(\theta^{(t-1)}, B_c)$
4: **end for**
5: $\Delta^{(t)} = \text{aggregate}(\Delta_c^{(t)})$      $\triangleright$ Compatible with SecAgg and DP
6: **return** $\theta^{(t)} = \text{server-optimizer}(\theta^{(t-1)}, \Delta^{(t)})$

7: **function** CLIENTUPDATE$(\omega^0, B_c)$
8:      **for** $k = 1, \ldots, N$ **do**
9:          Sample a minibatch $G \subset B_c$
10:          Compute gradient $g = \frac{\partial}{\partial \theta} \sum_{(x,a,r,\rho) \in G} \frac{1}{2}(f(x, a) - r)^2$      $\triangleright$ Regression-based loss
11:          Compute gradient $g = \frac{\partial}{\partial \theta} \sum_{(x,a,r,\rho) \in G} \frac{1}{2\rho}(f(x, a) - r)^2$      $\triangleright$ Importance weighting loss
12:          $\omega^k = \text{client-optimizer}(\omega^{k-1}, g)$
13:      **end for**
14:      **return** $\Delta_c^{(t)} \leftarrow \omega^N - \omega^0$
15: **end function**

---

In this section, we describe the federated CB algorithms that are developed and studied in this paper. The federated CB algorithms that we design are federated versions of the centralized CB algorithms described in Section 2.2. Recalling the general framework of Algorithm 1, we consider a meta iterator in the outer-loop named period, which can possibly run forever in an online setting, i.e., $I = \infty$. Each period simulates the deployment of a machine learning model parameterized by some parameters $\theta_{i-1}$, which can be less frequent for on-device applications compared to a web service. We focus on regression-based CB algorithms as in

Section 2.2, where the parameters $\theta$ induce a regression model which predicts the expected reward of actions $a$, given context $x$. Each period $i$ consists of some number of bandit inference steps followed by a training. At the beginning of each period, an inference model is deployed to all clients, and the model is trained with bandits data generated by a (delayed) inference model from the last period. For simplicity of presentation, we use the same number of examples at each client in inference, and do not incorporate heterogeneous delays in model deployment across clients as mentioned before.

Algorithm 2 describes the details of the inference procedure that happens asynchronously at each client. Client $c$ observes a context $x \sim D_c$. Given the current model parameters $\theta = \theta_{i-1}$, we use $f_\theta$ to refer to the induced reward predictor. This reward predictor $f_\theta$ is used to define a probability distribution over the actions as described in lines 4-6. The Greedy strategy is implemented by setting $\epsilon = 0$ in $\epsilon$-Greedy. The chosen action $a$ is subsequently drawn from this probability distribution, and the observed reward is logged along with the context, action and sampling probability in a local data log $B_c$ (line 8). In practice, where the number of inference examples handled at a client is exogenously determined, each client observes a potentially different number of inference examples in a period, $B_c$ is maintained locally on client and can be configured with suitable cap $M$ on the size of the local data log to respect memory and system constraints. Local cache $B_c$ potentially contains inference examples predicted by multiple previous model $\theta_0, \ldots, \theta_{i-1}$ due to heterogeneous delays of model deployment. When the deployment period is large, most of the clients participate in training contain lcoal cache of examples predicted by the most recent inference model $\theta_{i-1}$, and hence we reset $B_c$ every round for simplicity in simulation when used Algorithm 4.

Next, we discuss the algorithmic details of the training period, described in Algorithm 3. At a high-level, this procedure boils down to identifying an appropriate optimization objective on the local data logs of all the clients, which can then be optimized by any standard federated optimization algorithm. We consider two optimization objectives, motivated by the two predominant algorithmic settings in centralized CB. We describe their expected versions here, with the understanding that actual implementations use sample averages. The simplest objective is a regression on observed rewards as described before (Agarwal et al., 2012; Foster & Rakhlin, 2020; Simchi-Levi & Xu, 2022):

$$\text{Regression:} \quad \min_{f \in \mathcal{F}} \mathbb{E}_{c \sim p} \sum_{(x,a,r,\rho) \in B_c} (f(x,a) - r)^2. \tag{5}$$

When the class $\mathcal{F}$ is rich enough to satisfy $\mathbb{E}[r|x,a] \in \mathcal{F}$, this objective is natural, as the minimizer converges to the true expected rewards. However, if this assumption is grossly violated, then the regression objective can learn an unreliable predictor. A potentially preferable objective in such contexts is the following importance weighted regression variant (Bietti et al., 2021):

$$\text{Importance-weighted regression:} \quad \min_{f \in \mathcal{F}} \mathbb{E}_{c \sim p} \sum_{(x,a,r,\rho) \in B_c} \frac{1}{\rho} (f(x,a) - r)^2, \tag{6}$$

where $\rho$ is the recorded probability of choosing $a$ given $x$ in the local data log. Importance-weighting ensures that the objective is an unbiased estimator of $\mathbb{E}_{c \sim p} \mathbb{E}_{(x,r) \sim D_c} \sum_{a \in \mathcal{A}} (f(x,a) - r)^2$, so that the learned reward estimator is uniformly good for all the actions. This leads to strong guarantees for any function class $\mathcal{F}$, at the cost of a harder to optimize and higher variance training objective. We note that the application of FALCON with importance weighted updates is not considered in the original paper (Simchi-Levi & Xu, 2022). For our experiments, we primarily focus on the regression version as it displays superior empirical performance.

For either objective, we note that the underlying optimization problem clearly fits the form of the standard federated learning objective (1), meaning that off-the-shelf federated optimization algorithms can be readily applied. Federated Averaging (FedAvg) (McMahan et al., 2017) is a popular choice in pracitce, as it achieves both communication efficiency and fast convergence under heterogeneity (Wang et al., 2022). In Algorithm 3, we adopt the generalized FedAvg algorithm (Reddi et al., 2021; Wang et al., 2021), which views FL algorithms as two stage optimization: clients perform local training to compute model update $\Delta_c$, and the server uses

the averaged $\Delta$ as a pseudo gradient to update the global model $\theta$. The server performs such updates for $T$ rounds, sampling a fresh subset of clients at each round. Subsequently, the updated parameters are communicated to the clients for bandits inference, as mentioned earlier.

The updates on client and server require the specification of optimizers to be used. We follow standard practice and use stochastic gradient descent (SGD) as the client-optimizer as it works well and incurs no additional memory or computation overhead. We use Adam (Kingma & Ba, 2014) as the server-optimizer following Reddi et al. (2021).

**Differential privacy (DP).** The privacy properties of Algorithm 3 can be further improved via techniques like secure aggregation (Bonawitz et al., 2017) for the model updates, and by replacing FedAvg with variants that offer differential privacy (McMahan et al., 2018; Kairouz et al., 2021a; Choquette-Choo et al., 2022). We apply adaptive clipping (Andrew et al., 2021) with zero noise in aggregation as this has been shown to improve robustness with minimal computation and communication cost (Charles et al., 2021) in the bulk of our evaluation. In some of our experiments, we show the easy composition with differential privacy by introducing two additional operations for DP-FedAvg (McMahan et al., 2018): clip the model update $\widetilde{\Delta}_c^{(t)} = \min\left(1, \frac{C}{||\Delta_c^{(t)}||}\right)\Delta_c^{(t)}$ with clip norm $C$ estimated by adaptive clipping (Andrew et al., 2021); add Gaussian noise with standard deviation $\sigma C$ to $\Delta^{(t)} = \text{aggregate}(\widetilde{\Delta}_c^{(t)})$, where $\sigma$ is noise multiplier and $C$ is the clip norm.

## 4    Simulation Setup

In this section, we describe the setup used for our simulations of real-world federated CB problems. We describe the datasets used in our simulation, a detailed specification of the algorithms in the simulation setting, and the various settings that we simulate. Our code will be open-sourced.

### 4.1    Datasets for Simulating Federated CB

The methods that we evaluate roughly correspond to those outlined in Sections 2.2 and 3. Concretely, we evaluate the `Greedy`, $\epsilon$-`Greedy`, `Softmax` and `FALCON` strategies described above. For each strategy, we consider a few choices of the hyperparameters and mainly show the results for the best choice in a particular experimental condition. Details of the hyperparameters used can be found in Appendix A.

We use two datasets two evaluate these methods across a range of simulation settings in this work. The datasets are `EMNIST` and `StackOverflow` (`SO`), both of which have been used in prior works on federated learning. `EMNIST` is a handwritten character recognition dataset, comprising of digits (0-9) and letter (a-z, A-Z) inducing a multi-class classification problem with 62 labels. The dataset consists of characters written by different individuals, which are mapped to the different clients in the federated setting. We use the EMNIST dataset of 3400 clients provided by Tensorflow Federated (TFF Authors, 2022a) to train a two-layer convolutional neural network (CNN) (McMahan et al., 2017; Reddi et al., 2021). In bandit interaction, the learner predicts a class label upon seeing a character, and only gets a feedback about the correctness of this prediction, but does not observe the ground-truth label when this prediction is wrong, following the setup from prior works (Dudík et al., 2014; Bietti et al., 2021).

`SO` (TFF Authors, 2022b) is a language dataset of processed question and answer text with additional metadata such as tags. The dataset contains 342,477 unique users as training clients. We consider the tag prediction task and use a linear model based on the bag of words features for the sentences in each client. A vocabulary of 10,000 most frequent words is used. To make exploration feasible, we limit the tag set to the 50 most frequent tags. The original tag prediction is a multi-label and multi-class classification problem, and similar to EMNIST in bandit interaction, the learner will only get feedback about the correctness of a single predicted tag without observing the ground-truth label.

Next we discuss the various simulation setups used in this work.

---

**Algorithm 4** Federated Contextual Bandits in Simulations

---

**Require:** Communication rounds $T$ per period; training periods $I$; initial inference model $\theta_0$; bandits inference algorithm and hyparameters in Algorithm 2; federated optimization algorithms and hyparameters in Algorithm 3.

1: **for** $t = 1, 2, \ldots, IT$ **do**
2:      $i = \lceil t/T \rceil$
3:      Send training model $\theta^{(t-1)}$, inference model $\theta_{i-1}$ from server to a subset clients $\mathcal{S}^{(t)}$
4:      **for** each $c \in \mathcal{S}^{(t)}$ **in parallel do**
5:          $B_c \leftarrow \text{BanditInference}(\pi, \theta_{i-1})$                             $\triangleright$ Algorithm 2
6:          $\Delta_c^{(t)} = \text{ClientUpdate}(\theta^{(t-1)}, B_c)$                      $\triangleright$ Algorithm 3
7:      **end for**
8:      $\Delta^{(t)} = \text{aggregate}(\Delta_c^{(t)})$
9:      $\theta^{(t)} = \text{server-optimizer}(\theta^{(t-1)}, \Delta^{(t)})$
10:      **if** $t \mod T == 0$ **then**
11:          $\theta_i \leftarrow \theta^{(t)}$
12:      **end if**
13: **end for**

---

## 4.2   Simulation Scenarios

We consider three simulation scenarios in this paper. They roughly correspond to the scenarios where the CB agent starts from scratch, as is typically assumed in theory, as well as two settings where it starts from an initial model pre-trained with supervised data from a small number of clients, before being deployed in the CB setting. In the first pre-training setting, the reward distribution is the same in the pre-training and deployment phases, while the second one considers a distribution shift on the rewards. We begin with the high-level details of mapping the abstract federated CB framework of Algorithm 1 into a simulation setting, before describing the 3 variants below.

**Simulated federated CB:** When simulating a federated CB problem from a supervised dataset like EMNIST or SO, we need to choose the inference and training periods. For simplicity, we consider each period $i$ in Algorithm 1 to consist of $T$ communication rounds in Algorithm 4, which contains detailed implementation of the simulation framework. In each round $t \in [T]$ of a period $i \in [I]$, we choose a subset $\mathcal{S}^{(t)}$ of the client population. This represents the clients which participate in federated training at round $t$ in period $i$ in Algorithms 1 and 3. We limit the inference to only happen at the clients selected for training at this round, since the inference data generated at the other clients does not matter for model updates. While the inference rewards at all the clients are needed for measuring the performance of the deployed model, the average over the selected clients provides an unbiased approximation and makes the simulation computationally more tractable. Upon generating the inference data log $B_c$ at all the selected clients $B_c$, we then perform $N$ local updates, followed by an aggregated update of the server parameters. Upon the completion of $T$ such rounds, the client parameters are updated at each client and a new period starts. In this manner, each client has parameters delayed by up to $T$ rounds relative to the server. Note that a minor mismatch between the descriptions of Algorithms 1 and 4 is that if a client is selected at two different rounds within a period, then it uses an identical data log $B_c$ at both the periods in Algorithm 1, but samples a fresh log $B_c$ in Algorithm 4.

Next we describe how the client distributions are simulated using the supervised learning datasets in multiple ways below.

**Training from scratch with no initial model (scratch)** This scenario is the closest to the federated generalization of the standard CB setting studied in most papers. The server and clients start with some

randomly initialized model $\theta_0$. The model is used to choose actions for the inference period. The rewards of chosen actions are based on the classification loss, namely 1 for the action corresponding to a correct label and 0 otherwise.

**Initial model on a subset of clients (`init`)**  This scenario roughly captures a common practice in the industry where some small number of users (say employees of the organization) might try the application before broader deployment. In such cases, these initial users might even supply a richer feedback on the algorithm's chosen actions, an extreme case of which is providing the ground-truth label on each example, which allows the instantiation of rewards of all the actions. We model this by selecting a small number of clients for pre-training, and use supervised learning to minimize the squared loss across all the actions for each $x$, given the full reward vector. With this initial model, we then deploy to a broader client pool. Subsequently, the model is again updated in every training period in the same manner as the `scratch` scenario. We choose the number of initial clients to be 100 for both `EMNIST` and `SO`.

**Initial model on a subset of clients with reward distribution shift (`init-shift`)**  In practice, it is often unrealistic to assume that the reward distribution for model pre-training will match that during deployment due to a number of factors. The distribution of best actions within a subset of initial users (such as within an organization) might be heavily skewed relative to the general population. If the supervision for the initial model is instead obtained by third-party labelers, then there can be a mismatch between their preferences and those of the users. Finally, even beyond these, most practical problems exhibit non-stationarity (Wang et al., 2021; Zhu et al., 2021; Wu et al., 2022; Jiang & Lin, 2022) due to seasonal or other periodic effects, drifts in user preferences over time, changes in the action set etc. For example, emoji and users' preference can gradually change in an emoji recommendation application (Ramaswamy et al., 2019). In a way, some distributional mismatch between initial and deployment phases is likely most representative of the current practical scenario, and we treat this as our *default scenario.*

In `EMNIST`, we simulate this distribution shift by setting the reward in the initial training to be 1 if the label is correct, 0.5 if the true label is an upper-case letter and we predict its lower-case version and 0 otherwise. During the subsequent bandit simulation, we use the 0-1 valued rewards for exact match with the label, causing a label distribution shift.

In `SO`, we model distribution shift from two sources. The initial training only gets a multilabel 0/1 feedback based on tags in the 10 most frequent tags. That is, the learner sees a vector of labels of size 10, which has value 1 for all the tags in which are present in the example and 0 otherwise. However, the tag-set is expanded to the top 50 tags in the deployment phase, where the reward of a tag is defined as inversely proportional to the frequency of the tag in the corpus. Thus, the algorithm gets a higher reward for correctly predicting rare tags, which are not likely to be observed in the pre-training phase.

**Simulation durations**  Throughout the experiments, we use a total of 800 communication rounds (corresponding to $IT$ in Algorithm 4) for `EMNIST` and 1600 communication rounds for the larger `SO` benchmark, and randomly sample 64 clients in each round. The number of training periods $T$ is set to 4 for `EMNIST` and 8 for `SO` unless otherwise specified, corresponding to the deployment of a new model every 200 communication rounds. For `init` and `init-shift`, where we train an initial model for 100 iterations of supervised training, we only perform 700 (respectively 1500) rounds in the bandit phase for `EMNIST` (respectively `SO`). The comparison across settings at the end of training is not completely fair, however, as 100 rounds of supervised training provide significantly more information than 100 rounds of bandit interactions, since we observe feedback on all the actions in the supervised setup. We note that the scale of rewards also changes due to the rewards configuration in the `init-shift` setting.

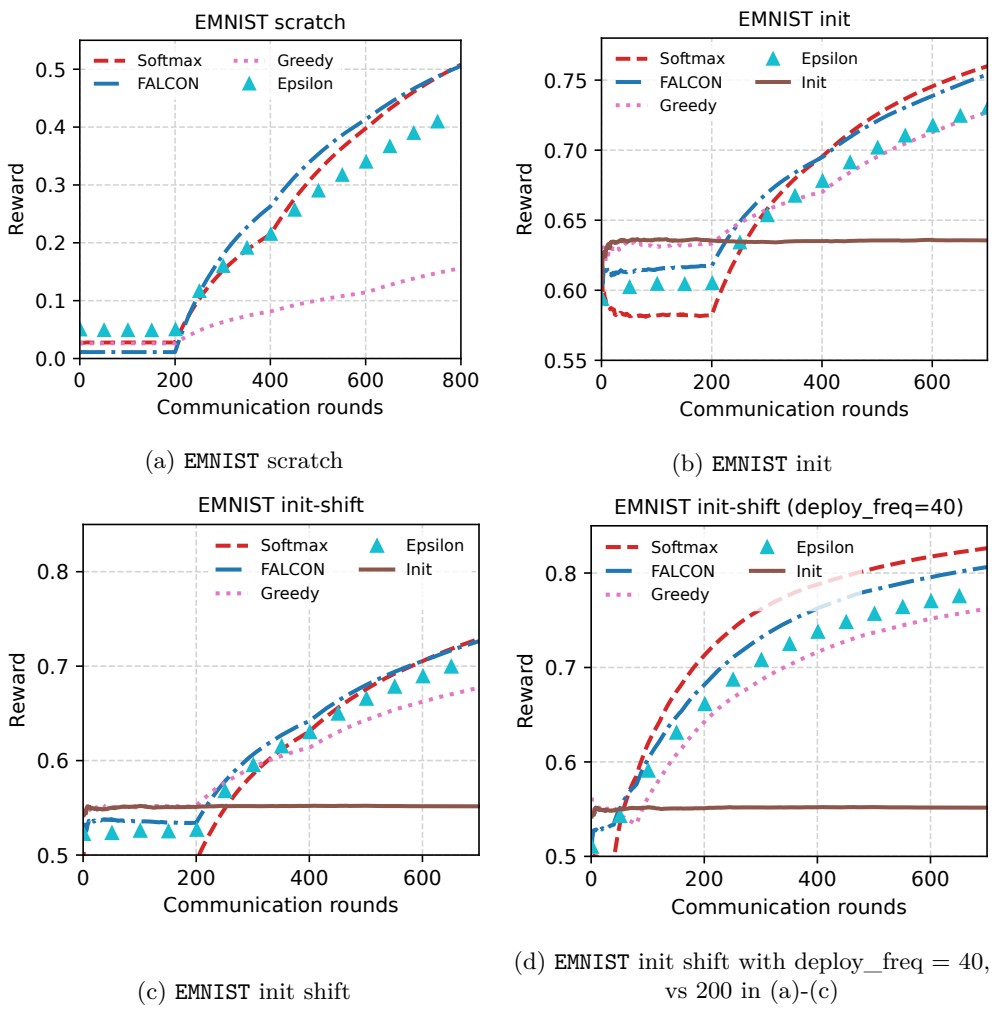

(a) EMNIST scratch

(b) EMNIST init

(c) EMNIST init shift

(d) EMNIST init shift with deploy_freq = 40, vs 200 in (a)-(c)

Figure 1: EMNIST experiments, without importance weighting. The *y*-axis gives *running average* reward, with different scales for each plot. While the regression model is the same for the first 200 rounds of each scenario, cumulative rewards are different depending on the amount of exploration done by the policy. The "Init" lines correspond to the greedy policy on the initial model, with no additional training. All the plots use the exploration parameters $\beta = 0.05$ and $\epsilon = 0.05$ for Softmax and $\epsilon$-Greedy respectively. Learning rate and exploration parameter values for each algorithm are detailed in Tables 1-4 for Figures 1a-1d respectively.

## 5 Empirical Evaluation Results

We begin with an evaluation of the baselines mentioned in the previous section across all the different experimental settings, before studying the effect of changing some important aspects of the setup as well as algorithmic choices.

### 5.1 Results for the three simulation settings

In Figures 1 and 2, we show a comparison of the different bandit algorithms on the EMNIST and SO benchmarks, respectively, across a range of experimental settings. In most of the experiments, we deploy a new

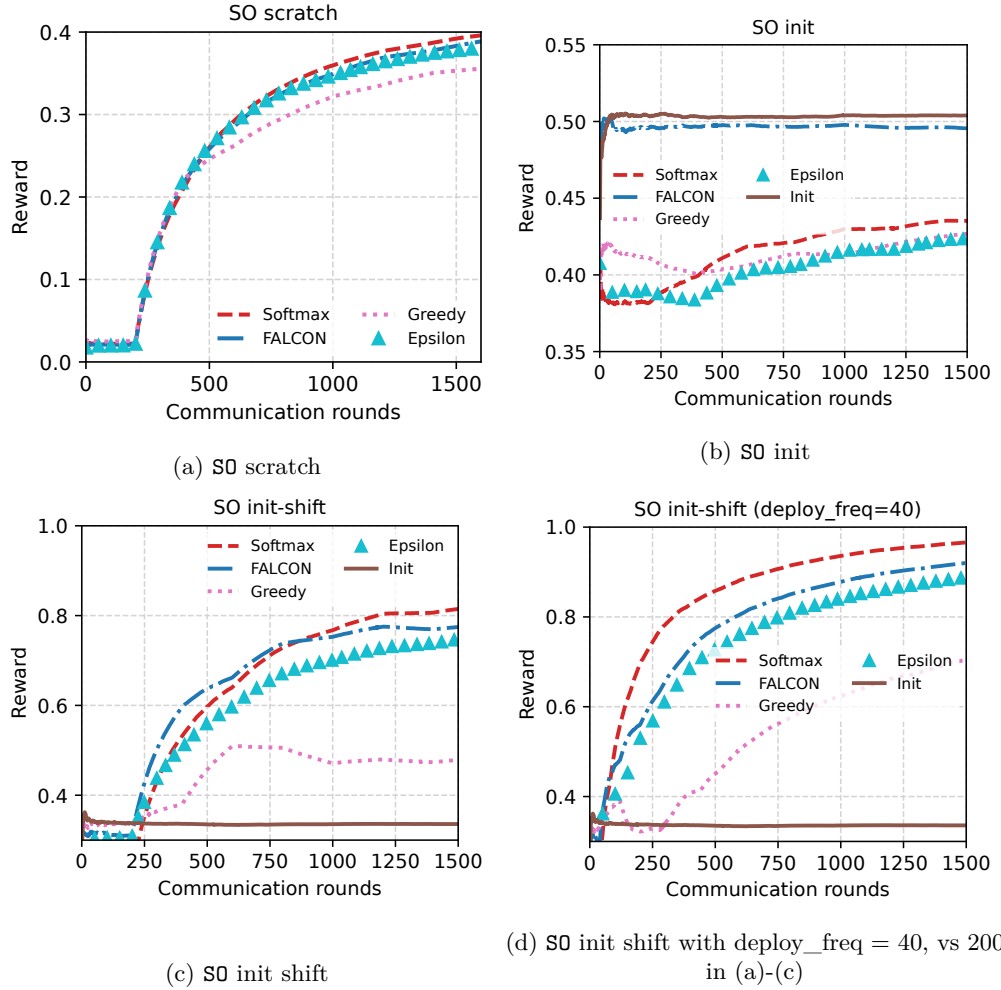

(a) SO scratch

(b) SO init

(c) SO init shift

(d) SO init shift with deploy_freq = 40, vs 200 in (a)-(c)

Figure 2: `StackOverflow` experiments. Note the different $y$-axis reward scales on the different plots. Learning rate and exploration parameter values for each algorithm are detailed in Tables 5-8 for Figures 2a-2d respectively.

model every 200 communication rounds, while the settings vary in {`scratch`, `init`, `init-shift`}, with the results summarized in Figures 1a-1c and 2a-2c for the two benchmarks.

As a first takeaway, we note that *exploration almost always helps* relative to the baseline `Greedy` strategy, and never hurts, even as the extent of gains can be dependent on the setting. When starting without an initial model in the **scratch setting**, exploration is typically crucial since the initial model can arbitrarily prefer certain actions. This is most clearly reflected in Figure 1a for the `EMNIST` benchmark, although the absolute reward is quite low in both `EMNIST` and `SO` at the end of the experiment in both the cases for this setting, meaning that the regime might be less relevant practically. While exploration is generally helpful, it is critical to balance the explore-exploit tradeoff, and best performance is generally achieved for parameter settings that result in fairly aggressive exploration early on, before converging closer to a greedy choice towards the end of training in both `FALCON` and `Softmax` algorithms. In Section 5.3, we quantify this phenomenon for `Softmax` in Figs. 4b and 4d while also showing noise added for differential privacy also has an effect.

In the **init setting**, the results are more mixed since the algorithms start with an initial model which already has a strong performance. For instance, the initial model has a higher reward than the performance at the

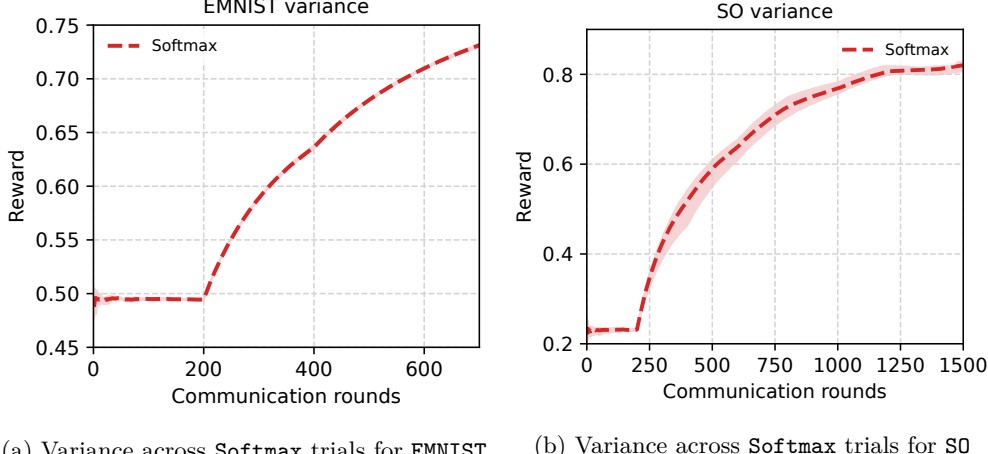

(a) Variance across `Softmax` trials for `EMNIST`

(b) Variance across `Softmax` trials for `SO`

Figure 3: Variance across 5 trials of `Softmax` in the `init-shift` setting for `EMNIST` and `SO`

end of training from scratch in both Figures 1b and 2b. Consequently, there is little benefit from additional learning, and we find that the best results are attained for hyperparameters that favor little exploration, and small optimization updates through small learning rates. This is also reflected in the nearly identical behavior as `Greedy` for most exploration strategies other than `FALCON` for `SO` in Figure 2b. We expect that the performance of `Greedy` deteriorates with respect to the initial model, because a smaller learning rates close to zero outside our search grid can be preferable when initial model is very strong. Nevertheless, the overarching conclusion we draw here is that even small amounts of high quality *fully supervised* data can be very powerful, when the downstream model does not encounter any subsequent distribution shift.

Expecting stationarity after deployment, or fully representative labeled set in training the initial model, however, is an unrealistic assumption, which is the reason we focus on the **init-shift setting** as our primary one. Here, we again find that *exploration helps substantially*, and the preferred hyperparameters result in more aggressive exploration as well as larger optimization steps. This is particularly pronounced in Figure 2c, where the initial model is quite poor, `Greedy` gets a middling improvement on it while the exploration algorithms all reach significantly larger rewards. For Figure 1c, the preferred exploration parameters are comparitively less aggressive, and this is also reflected in a smaller edge over `Greedy`. Overall, this reinforces the intuition that some amount of persistent exploration is beneficial in dynamic, non-stationary environments.

Given this evaluation across settings and algorithms, we are ready to present the first high-level takeaway from our experiments for practitioners:

> **Takeaway 1: Effectiveness of `Softmax`.**
>
> *We find that the `Softmax` approach, while being a simple modification of the `Greedy` strategy, has a re-markably strong performance across benchmarks and experimental settings, always either performing the best or close to it. While `FALCON` performs comparably well, the fact that getting strong explo-ration performance requires tuning two unrelated hyperparameters is a serious practical drawback. Consequently, we recommend `Softmax` as an effective default strategy for practitioners.*

**Variance across repeated trials.** All our algorithms are randomized due to the random sampling in-volved in exploration. The simulation itself has many random choices such as the choice of which clients participate in a training round and example selection in each mini-batch. The conclusions discussed so far

are remarkably robust to this randomness, and we show the stability in our results for the recommended `Softmax` strategy in the `init-shift` setting in Figure 3. As we see, the variation in rewards across repeated trials is negligible.

### 5.2 A Closer look at some choices in the algorithms and setup

We now take a deeper look into some of the choices both in our setup and the design and implementation of the algorithms which can lead to a significant change in the results, and hence are important to be aware of in practice. We start with a common practical question of the effect of model deployment frequency, corresponding to the number of model updates and training rounds that the algorithm faces.

**Effect of deployment frequency.** So far, we have discussed results where new models are deployed once every 200 communication rounds. The choice of deployment frequency is itself a tunable parameter in practice, although very small frequencies are typically infeasible from system considerations, and often undesirable from a stability perspective. In Figures 1d and 2d, we investigate the performance of algorithms in the `init-shift` setting, when the deployment frequency is reduced to just 40 rounds. This means that we get a total of 20 training periods in `EMNIST` and 40 periods in `SO`. The first observation is that the absolute performance of all the methods improves over the corresponding Figures 1c and 2c with a frequency of 200 in the same setting. This is not surprising as better models are deployed early with a smaller deployment frequency, giving a longer time to effectively exploit the gains from exploration. This confirms the intuition that smaller deployment frequencies are preferable from a learning perspective, as long as the rest of the system architecture allows it.

Next we study the effect of varying some important elements in Algorithm 4.

**Effect of optimizer choice.** Algorithm 3 allows us to choose different client and server optimizers. We fix client optimizer to SGD throuhgout, but use ADAM (Kingma & Ba, 2014) as the default choice for server optimizer, consistent with prior works on supervised federated learning (Reddi et al., 2021). We test the effectiveness of this choice by changing the server optimizer to SGD for `Softmax` in the `init-shift` setting in both `SO` and `EMNIST`. While there is no change in the final performance at tuned hyperparameters for `EMNIST`, the average bandit reward at the end of 1500 communication rounds drops from 0.81 to 0.62. This mirrors prior results in the supervised setting (Reddi et al., 2021), where ADAM is found to be superior for the `SO` task, due to the presence of sparse, high-dimensional features.

**Effect of importance sampling.** As we discuss in Section 3.2, several prior works train an importance weighted regressor (Bietti et al., 2021) to form the underlying greedy policy in $\epsilon$-`Greedy`, while we adopt an unweighted regression. This is due to the destabilizing effects of the variance from importance weighting on the learning process. Indeed, we find that changing the $\epsilon$-`Greedy` approach to use weighted regression worsens the performance in the `init-shift` setting from 0.71 to 0.6 for `EMNIST` and from 0.72 to 0.47 for `SO`. There is a wealth of literature on variance reduction techniques with importance weighting, such as the doubly robust methods (Dudík et al., 2014). However, given the strong performance of unweighted methods here, we do not investigate these additional techniques due to added challenges with hyperparameters and learning complexity in practice. While the theoretical foundations of the unweighted approach here rely on an expressivity assumption on the underlying function class as we discuss in the next section, we find that this is less of a concern in modern systems with powerful, over-parameterized regression models.

> **Takeaway 2: Importance of variance control.**
>
> *Both the choice of ADAM versus SGD as server optimizer and the use or not of importance weights eventually control the variance in the training process, and crucially modulate the sample efficiency in our experiments. We find the choices of ADAM and regression-based loss to be effective across settings, and recommend them to practitioners.*

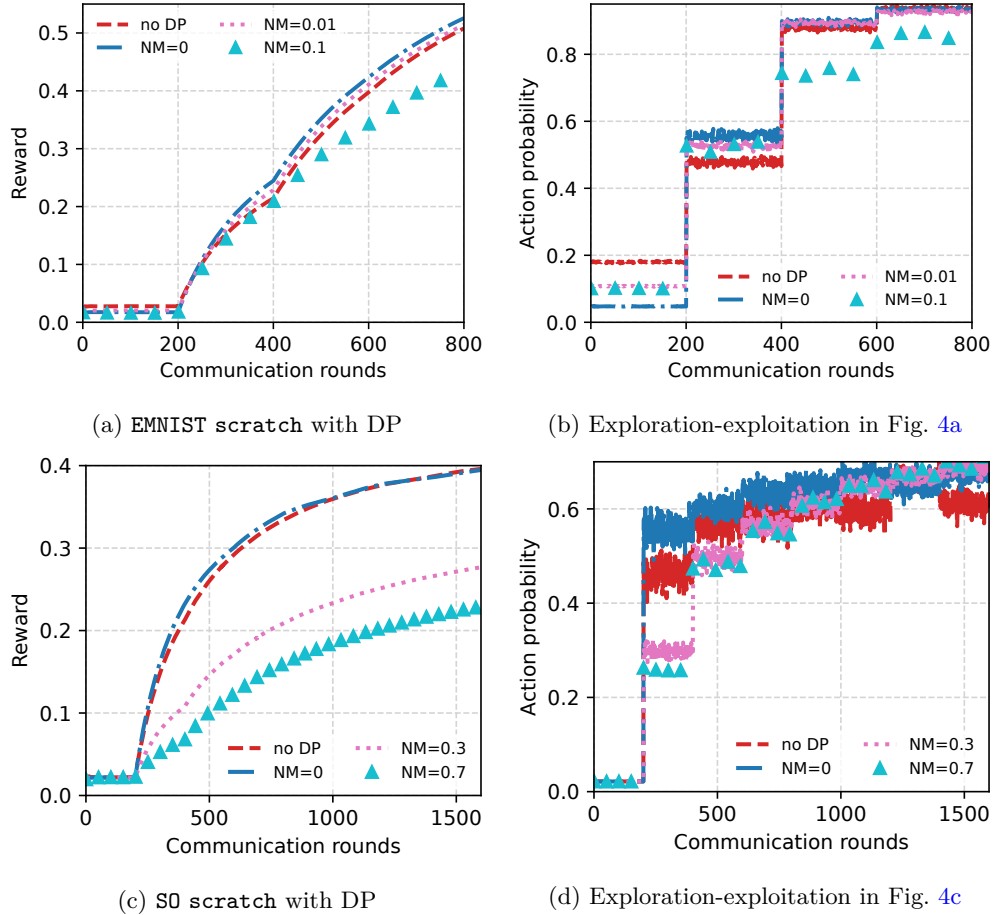

(a) EMNIST scratch with DP

(b) Exploration-exploitation in Fig. 4a

(c) SO scratch with DP

(d) Exploration-exploitation in Fig. 4c

Figure 4: Differential privacy for Softmax variations in the scratch setting. Hyperparameters are detailed in Table 9

**Choosing hyperparameters.** While hyperparameter choice is a process fraught with some overhead in all learning pipelines, it is particularly challenging in bandit settings, where each hyperparameter drives different data collection and hence tuning is not so straightforward. Unfortunately, we find that while the exploration parameters show remarkable stability for most approaches and regimes, the optimization learning rates are more sensitive. For Softmax, a temperature parameter of 0.05 performs the best in all regimes other than init-shift, where a slightly higher choice of 0.1 does somewhat better, though 0.05 is still reasonably good. Similarly $\epsilon = 0.05$ works best in most cases for $\epsilon$-Greedy. FALCON, in contrast, requires very different choices across datasets and settings, explaining our preference of Softmax over FALCON. For optimization parameters, we find that higher learning rates are preferred in scratch and init-shift settings, while init prefers smaller learning rates due to the high-quality initial model. Since practical setups typically use fairly large deployment frequencies, it is reasonable to pick the optimization hyperparameters through offline off-policy evaluation style approaches (Dudík et al., 2014) from the accumulated training data. See Appendix A for hyperparameter tuning details.

### 5.3 Incorporating differential privacy.

We provide preliminary results on adding differential privacy to the federated CB experiments by applying DP-FedAvg (McMahan et al., 2018) in Algorithm 3, as discussed in Section 3. We consider the scratch

setting in Fig. 4, but same approach can be applied in the `init` and `init-shift` settings after accounting the privacy budget for pretraining or pretraining on public data. We follow the strategy in Xu et al. (2022) to tune the hyperparameters: we first estimate an (aggressive) clip norm with adaptive clipping (Andrew et al., 2021) of target quantile 0.5 and noise multiplier 0, and a small grid of learning rates around the best learning rates tuned in no DP settings; we fix the clip norm to 0.1 for `EMNIST` and 0.8 for `StackOverflow` and then choose a small and large noise multiplier respectively for `EMNIST` and `SO`; we further tune the learning rates in a small grid based on the learning rates chosen for adaptive clipping experiments, and select the best hyperparameters based on the final (averaged) reward.

Fig. 4 compares four approaches:

- **No DP** shows vanilla FedAvg with adaptive clipping of to a large target quantile (0.8) that clips rarely, without noise.

- **NM=0** uses a fixed clipping norm with no added noise.

- **NM=0.01 or 0.3**, small noise multipliers for `EMNIST` and `SO` respectively.

- **NM=0.1 or 0.7**, corresponding large noise multipliers.

The large noise multiplier will conceptually result in stronger privacy guarantees, however, for the small `EMNIST` dataset of 3400 clients, even NM=0.1 is not large enough to achieve meaningful formal DP guarantees. For `SO` of $\sim$0.34M clients, when assuming Poisson sampling and add-or-remove-one adjacency, we use RDP (Abadi et al., 2016; Mironov, 2017) accounting to compute privacy guarantees measured by $(\epsilon, \delta)$-DP. Fixing $\delta = 10^{-6}$, the noise multipliers 0.3 and 0.7 can lead to $\epsilon = 15.8$ and 1.5 respectively.

Fig. 4a (`EMNIST`) and Fig. 4c (`SO`) show the running average reward of these approaches, and suggest that clipping alone does not necessarily degrade the model utility measured by reward, and noise multiplier controls the privacy-utility trade-off. The observations of the DP effect in bandits settings are similar to the previous observation in supervised settings (Andrew et al., 2021; Kairouz et al., 2021a). The preliminary DP results are provided to show the proposed federated bandit algorithms are indeed compatible with differential privacy. There are many potential tuning strategies to achieve stronger privacy-utility trade-offs (Ponomareva et al., 2023). A particular useful tuning strategy for DP is to sample large number of clients per round. Following (McMahan et al., 2018; Kairouz et al., 2021a; Xu et al., 2022), we can extrapolate the privacy and utility in a more realistic setting by assuming larger number of total clients, and linearly increasing noise multiplier and clients per round. Figure 4a shows that **NM=0.01** can achieve strong utility. When $\sim 0.34$M total clients can participate in training, and scaling up NM from 0.01 to 1, RDP accounting can achieve $(\epsilon = 4.13, \delta = 10^{-6})$-DP. If we also linearly scale up clients per round from 64 to 6400, the utility measured by reward is expected to be similar to the strong utility of **NM=0.01** in Fig. 4a.

In Figs. 4b and 4d, we further report the probability of the chosen action ($p_j(a^j)$ in Algorithm 2) averaged for data of the sampled clients in each round, which is an indicator of the exploration-exploitation trade-off of the `Softmax` algorithm. The `Softmax` algorithm has an interesting annealing effect of exploration: the probability of chosen actions gradually increase as the models become more confident in their predictions after training. DP seems to have a larger effect on the probability at the early stage of training for `SO`, while the effect happens at later stage for `EMNIST`. The relationship of randomness in bandits exploration and the noise addition of DP can be an interesting topic for future study.

## 6   A theoretical model

We now present a simple theoretical model to understand some of the key considerations in federated CB learning. Using the same high-level setup as Section 3.1, we abstract the inference and training periods as described below.

**Inference:** At inference period $i$, each client $c$ simultaneously uses the currently available model $\pi_i$ to choose actions for any contexts $x \sim D_c$ that it observes, and logs $(x, a, \mathfrak{r}, \pi_i(a|x))$, with $a \sim \pi_i(\cdot|x)$ and $\mathfrak{r} = r(x, a)$ for $(x, r) \sim D_c$.

**Training:** At each training period $i = 1, 2, \ldots$, the server updates the model using a total of $n$ new training log entries for this training period, distributed across the clients participating in the training period. To abstract away the specifics of client sampling and its effects, we consider the $n$ samples to be i.i.d. according to the choice of a client $c \sim p$ and $(x, r) \sim D_c$.

We make an additional assumption on the problem setup which leads to computationally nicer algorithms. Concretely, we assume that our CB algorithm models the rewards, and has access to a function class $\mathcal{F} \subseteq \{\mathcal{X} \times \mathcal{A} \to [0, 1]\}$, so that each $f \in \mathcal{F}$ predicts rewards, given a context, action pair as the input. To obtain theoretical justification for the use of such a parameterization, centralized CB algorithms make the so-called *realizability assumption* that for some $f^\star \in \mathcal{F}$, $\mathbb{E}[r|x, a] = f^\star(x, a)$ for all $x, a$. However, in the federated setting, we have heterogeneous data distributions across clients. Nevertheless, we use a common set of parameters to predict the rewards at each client, which motivates the following realizability assumption in the federated setting.

**Assumption 1** (Realizability in Federated CBs). *There exists $f^\star \in \mathcal{F}$ such that $\mathbb{E}_{D_c}[r|x, a] = f^\star(x, a)$ for all $x \in \mathcal{X}, a \in \mathcal{A}$ and $c \in \mathcal{C}$.*

Importantly, this assumption does not contradict the substantial heterogeneity in client preferences that may naturally arise in federated settings, as such heterogeneity can be modeled via appropriate distributions $D_c$, allowing a single $f^\star$ to effectively behave arbitrarily differently on different clients (e.g., in the extreme case where the support of the clients $D_c$ is non-overlapping).

Under the realizability assumption, it is natural to learn the regression function using the unweighted regression objective (5). To abstract away the details of the underlying FL algorithms, we assume access to a federated regression oracle which can optimize such objectives, formally:

**Definition 1** (Federated Regression Oracle). *Given clients $c_1, \ldots, c_m$ with local datasets $S_1^c, S_2^c, \ldots, S_m^c$ satisfying $|S_1^c \cup S_2^c \cup S_m^c| = n$ , a federated regression oracle returns a function $\hat{f}$, using a federated learning protocol, which satisfies:*

$$\frac{1}{n} \sum_{i=1}^{m} \sum_{(x,a,\mathfrak{r}) \in S_m^c} (\hat{f}(x, a) - \mathfrak{r})^2 \leq \frac{1}{n} \min_{f \in \mathcal{F}} \sum_{i=1}^{m} \sum_{(x,a,\mathfrak{r}) \in S_m^c} (f(x, a) - r)^2 + \epsilon_{\text{fed-opt}}.$$

The parameter $\epsilon_{\text{fed-opt}}$ captures the accuracy of solving the regression problem over $n$ examples distributed over $m$ clients in a federated manner, and will in general depend on the choice of the federated learning method, settings of hyperparameters such as communication rounds, etc. We assume that the clients $c_1, \ldots, c_m$ are chosen i.i.d. from the underlying distribution $p$, and that the effective training set for the regression problem $S_1^c \cup S_2^c \cup S_m^c$ (which is never explicitly materialized in one place) is of a fixed size $n$, with samples i.i.d. from the ideal sampling distribution $c \sim p$ and $(x, r) \sim D_c$.

**Federated inference regret of $\epsilon$-Greedy** With this background, it is straightforward to analyze a simple regression-based $\epsilon$-Greedy method for the federated setting. Let $\hat{f}_{i+1}$ be the regressor computed at the training period $i$. Furthermore, for any $f \in \mathcal{F}$, let $\pi_f(x) = \text{argmax}_a f(x, a)$ denote the greedy policy, with ties broken in an arbitrary manner, and let $\pi_i(x) = (1 - \epsilon)\pi_{f_i}(x) + \epsilon\text{Unif}(\mathcal{A})$ denote the inference policy deployed at inference period $i$ and $\pi^\star = \pi_{f^\star}$ denote the optimal policy. Since $f, r$ are both bounded in $[0, 1]$ and we use $n$ fresh training samples at each training period $i$ to have a total of $ni$ samples after $i$ periods, it can be show that (see e.g. (Agarwal et al., 2012)) with probability at least $1 - \delta$, the following generalization

bound for the regression performance of $\hat{f}_{u+1}$ holds:

$$\frac{1}{i}\sum_{j=1}^{i}\mathbb{E}_j\left[(\hat{f}_{i+1}(x,a)-r)^2-(f^\star(x,a)-r)^2\right]=\mathcal{O}\left(\frac{\ln(|\mathcal{F}|/\delta)}{ni}+\epsilon_{\text{fed-opt}}\right). \tag{7}$$

Here we use $\mathbb{E}_j$ as a shorthand to denote expectation over random variables $c\sim p, x, r\sim D$ and $a\sim\pi_j(\cdot|x)$. We also assume that the class $\mathcal{F}$ is finite for our analysis here for convenience. Using standard arguments, a similar result can also be obtained for infinite function classes through the use of covering. Under Assumption 1, the proof of Lemma 4.3 of Agarwal et al. (2012) further implies that

$$\mathbb{E}_{c\sim p}\mathbb{E}_{(x,r)\sim D_c}\left[r(\pi^\star(x))-r(\pi_{\hat{f}_{i+1}}(x))\right]\leq\sqrt{\mathbb{E}_{c\sim p}\mathbb{E}_{(x,r)\sim D_c}\left[\left(r(\pi^\star(x))-r(\pi_{\hat{f}_{i+1}}(x))\right)^2\right]}$$

$$\leq\sqrt{\frac{2K}{\epsilon}\frac{1}{i}\sum_{j=1}^{i}\mathbb{E}_j\left[(\hat{f}_{i+1}(x,a)-r)^2-(f^\star(x,a)-r)^2\right]}$$

$$=\mathcal{O}\left(\sqrt{\frac{2K}{\epsilon}\left(\frac{\ln(|\mathcal{F}|i/\delta)}{ni}+\epsilon_{\text{fed-opt}}\right)}\right), \tag{8}$$

where the first inequality follows from Jensen's inequality, the second inequality uses Lemma 4.3 of Agarwal et al. (2012), and in the last step we use Eq. (7). Since our actual inference policy $\pi_{i+1}$ is $\epsilon$-greedy, the per-round inference regret after $I$ training rounds is at most

$$\mathcal{O}\left(\epsilon+\sqrt{\frac{2K}{\epsilon}\left(\frac{\ln(|\mathcal{F}|I/\delta)}{nI}+\epsilon_{\text{fed-opt}}\right)}\right).$$

To better contrast this result with standard CB guarantees in the centralized setting, we make a simplifying assumption that we have only 1 client in the pool, and that the number of samples per inference period is the same as the size of our training pool for each period, equal to $n$. Then the cumulative inference regret after $I$ periods is at most

$$\left(\epsilon+\sqrt{\frac{2K}{\epsilon}\epsilon_{\text{fed-opt}}}\right)nI+n+\sum_{i=2}^{I}\sqrt{\frac{2K}{\epsilon}\cdot\frac{n\ln(|\mathcal{F}|I/\delta)}{(i-1)}}. \tag{9}$$

In comparison, under the same assumptions, updating the regressor after each inference round yields a regret of at most

$$\left(\epsilon+\sqrt{\frac{2K}{\epsilon}\epsilon_{\text{opt}}}\right)nI+1+\sum_{j=2}^{nI}\sqrt{\frac{2K}{\epsilon}\cdot\frac{\ln(|\mathcal{F}|nI/\delta)}{j-1}}, \tag{10}$$

where $\epsilon_{\text{opt}}$ is the accuracy of the centralized regression oracle. Assuming that the two optimization errors are of a comparable order, then the main difference in the two bounds arises due to the delay of roughly one inference period in the model updates in the federated setting. Clearly the gap is at most of a constant factor and decreases over time, which is consistent with prior results on delayed bandit learning. As we have already observed in the empirical evaluation, however, when the number of inference and training periods, given by $I$ above, is relatively small, then this delay has a non-trivial effect on the performance (see e.g. the effect of deployment frequency in Section 5.2). An extreme case of this can be observed by setting $I=1$, whence the bound in (9) becomes vacuously large in the final term, while that in (10) still decreases as $\widetilde{\mathcal{O}}(1/\sqrt{n})$ in the final term.

Note that our calculations above assume that our regression solution $\hat{f}_i$ fits all the training data accumulated over prior training periods $1, 2, \ldots, i-1$. In practice, depending on the implementation details, it might only

incorporate the data from the most recent, or roughly a constant number of past training periods, but where the optimizer is warm-started from the previous solution. As long as the optimizer does provide guarantees of approximately fitting the entire data through the warm-start however, our conclusions continue to hold in this setting.

**Federated inference regret of FALCON** . While our analysis of the $\epsilon$-greedy approach above serves to illustrate most of the key ideas and modifications in the federated setting from a centralized one, it has the drawback of a weak overall regret bound due to the simplistic uniform exploration. In the centralized setting, recent algorithms (Foster & Rakhlin, 2020; Simchi-Levi & Xu, 2022) have leveraged Assumption 1 to give statistically optimal CB results, and can be computationally implemented using regression oracles. For the federated setting, the FALCON algorithm of Simchi-Levi & Xu (2022) is particularly attractive, since it takes an offline squared loss regression oracle as an input, which can be instantiated with a federated regression oracle in the federated setting. This combination allows us to get a per-round inference regret after $I$ training rounds of

$$
\mathcal{O}\left(\sqrt{K\left(\frac{\ln(|\mathcal{F}|I/\delta)}{nI} + \epsilon_{\text{fed-opt}}\right)}\right),
$$

which removes the undesirable scaling of $\mathcal{O}(1/\epsilon)$ on a fixed exploration parameter through a more adaptive exploration-exploitation tradeoff. The effect of delays and other aspects of the comparison with the centralized setting remain unchanged.

## 7 Conclusion

This paper aims to provide a practical perspective on the important problem of federated contextual bandits, with a goal of both highlighting the relevance of this paradigm to real-world applications, and to demonstrate the effectiveness of simple strategies when instantiated with the right choices. An additional goal and contribution of this work is to develop a robust simulation methodology for the federated CB setting, which incorporates practical concerns such as leveraging small amounts of pre-training data, potentially mis-aligned with the eventual performance metrics, as well as non-stationarity and distributional shifts. Indeed some of these factors are rarely incorporated even in the most comprehensive centralized CB evaluation, and are of independent interest to the bandit community. For practitioners, we hope that the takeaways from our simulations on which algorithmic choices work well can be a useful guide to applying these ideas.

More generally, as we see an ever increasing focus on personalization and fine-tuning of large, general purpose models with RL, the availability of technologies such as federated CB and more general forms of federated RL are essential to our ability to learn in a private and responsible manner. Extending these ideas to more general forms of RL is an important direction for future work, as is a deeper understanding of the interplay between privacy and the RL setting.

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

## A  Detailed Hyperparameter Settings

We now give the detailed hyperparameter settings for the different simulation scenarios and algorithms.

Where we discuss choosing hyperparameters from a grid, unless otherwise noted we ran all combinations of the hyperparameters for each (scenario × algorithm × dataset) configuration, and report the runs which achieved the best running average reward at the end (last round) of training. As described in Algorithm 4, the same set of clients are used for bandit inference and federated training. For EMNIST, a client may be revisited $64 \times 800/3400 \sim 15$ times while for SO, as the dataset is large, it would be rare to revisit the same client twice.

### A.1  Settings for the exploration parameters

We begin with $\epsilon$-Greedy and Softmax, which use fixed hyperparameters across all simulations. The preferred choices which result in the highest CB reward at the end of the experiment are indicated in bold.

- $\epsilon$ for $\epsilon$-Greedy: $\epsilon \in \{\mathbf{0.05}, 0.1\}$.

- $\beta$ for Softmax: $\beta \in \{0.02, \mathbf{0.05}, 0.1\}$.

For the FALCON algorithm, we found setting the two hyperparameters of $\gamma$ and $\mu$ to be significantly more challenging. To have a standardized way of setting these across both the datasets, we first chose $\mu \in \{1, 0.1, 0.01\} K/\epsilon$ with $\epsilon = 0.05$, so that the contribution of this term is in various multiples of our preferred parameter in $\epsilon$-Greedy. Since the number of actions is different in the two cases, this results in $\mu \in \{12, 124, 1240\}$ and $\mu \in \{10, 100, 1000\}$ for EMNIST and SO respectively. For $\gamma$, we further tune it in the set $\gamma \in \{1000, 5000\}$, which we found to be reasonable for both the datasets.

### A.2  Settings for the optimization hyperparameters

Next we describe the optimization hyperparameters which are more sensitive to the dataset and the simulation setting used. We always choose learning rates from a grid of the form $\{1, 2, 5\} * 10^{-n}$, where $n$ is chosen appropriately for each setting. We used a fixed grid across algorithms and scenarios for each dataset, and when the best settings fell on the edge for an algorithm in a setting, we ran additional runs to confirm that expanding the grid does not improve the results. We start with the parameters for EMNIST.

- learning rate for client optimizer (SGD) $\in \{0.01, 0.02, 0.05, 0.1, 0.2, 0.5\}$.

- learning rate for server optimizer (ADAM) $\in \{0.0005, 0.001, 0.002, 0.005, 0.01, 0.02, 0.05\}$.

| Algorithm | Server learning rate | Client learning rate | Exploration param |
|---|---|---|---|
| Softmax | 0.002 | 0.1 | $\beta = 0.05$ |
| FALCON | 0.002 | 0.1 | $\mu = 12, \gamma = 1000$ |
| Greedy | 0.001 | 0.2 | · |
| $\epsilon$-Greedy | 0.01 | 0.1 | $\epsilon = 0.05$ |

Table 1: Hyperparameter settings for the EMNIST dataset and the scratch scenario (Fig. 1a)

| Algorithm | Server learning rate | Client learning rate | Exploration param |
|---|---|---|---|
| Init (supervised) | 0.5 | 0.5 | · |
| Softmax | 0.005 | 0.1 | $\beta = 0.05$ |
| FALCON | 0.002 | 0.2 | $\mu = 12, \gamma = 5000$ |
| Greedy | 0.002 | 0.1 | · |
| $\epsilon$-Greedy | 0.002 | 0.1 | $\epsilon = 0.05$ |

Table 2: Hyperparameter settings for the EMNIST dataset and the init scenario (Fig. 1b)

The corresponding settings for SO are:

- learning rate for client optimizer (SGD) $\in \{0.02, 0.05, 0.1, 0.2, 0.5, 1, 2, 5\}$.

- learning rate for server optimizer (ADAM) $\in \{0.0002, 0.0005, 0.001, 0.002, 0.005, 0.01, 0.02, 0.05\}$

We use the default values in Keras for the remaining ADAM hyperparameters such as $\beta_1, \beta_2$ and $\epsilon$. The large grids for the server optimizer are primarily because the init setting prefers a much smaller learning rate at the server than the other settings.

We fix other federated optimization parameters in all experiments: each client run one epoch on their local logged data for training; minibatch size of 16 is used on clients; 64 clients are sampled per round; the maximum number of samples per client on SO is capped at 256.

We conclude this section by giving tables of learning rate settings for each of the plots in Figures 1, 2 and 4.

| Algorithm | Server learning rate | Client learning rate | Exploration param |
|---|---|---|---|
| Init (supervised) | 0.5 | 0.5 | · |
| Softmax | 0.005 | 0.1 | $\beta = 0.05$ |
| FALCON | 0.005 | 0.2 | $\mu = 12, \gamma = 5000$ |
| Greedy | 0.001 | 0.1 | · |
| $\epsilon$-Greedy | 0.01 | 0.1 | $\epsilon = 0.05$ |

Table 3: Hyperparameter settings for the EMNIST dataset and the init-shift scenario (Fig. 1c)

| Algorithm | Server learning rate | Client learning rate | Exploration param |
|---|---|---|---|
| Init (supervised) | 0.5 | 0.5 | · |
| Softmax | 0.005 | 0.2 | $\beta = 0.05$ |
| FALCON | 0.005 | 0.1 | $\mu = 12, \gamma = 5000$ |
| Greedy | 0.002 | 0.1 | · |
| $\epsilon$-Greedy | 0.005 | 0.2 | $\epsilon = 0.05$ |

Table 4: Hyperparameter settings for the EMNIST dataset and the init-shift scenario with deploy_freq = 40 (Fig. 1d)

| Algorithm | Server learning rate | Client learning rate | Exploration param |
|---|---|---|---|
| Softmax | 0.01 | 1 | $\beta = 0.05$ |
| FALCON | 0.01 | 0.05 | $\mu = 10, \gamma = 5000$ |
| Greedy | 0.01 | 0.1 | · |
| $\epsilon$-Greedy | 0.01 | 0.2 | $\epsilon = 0.05$ |

Table 5: Hyperparameter settings for the SO dataset and the scratch scenario (Fig. 2a)

| Algorithm | Server learning rate | Client learning rate | Exploration param |
|---|---|---|---|
| Init (supervised) | 0.05 | 0.2 | · |
| Softmax | 0.005 | 0.05 | $\beta = 0.05$ |
| FALCON | 0.0005 | 0.1 | $\mu = 10, \gamma = 5000$ |
| Greedy | 0.001 | 0.02 | · |
| $\epsilon$-Greedy | 0.005 | 0.1 | $\epsilon = 0.05$ |

Table 6: Hyperparameter settings for the SO dataset and the init scenario (Fig. 2b)

| Algorithm | Server learning rate | Client learning rate | Exploration param |
|---|---|---|---|
| Init (supervised) | 0.05 | 0.05 | · |
| Softmax | 0.02 | 2 | $\beta = 0.1$ |
| FALCON | 0.05 | 0.2 | $\mu = 100, \gamma = 1000$ |
| Greedy | 0.05 | 1 | · |
| $\epsilon$-Greedy | 0.05 | 0.2 | $\epsilon = 0.05$ |

Table 7: Hyperparameter settings for the SO dataset and the init-shift scenario (Fig. 2c)

| Algorithm | Server learning rate | Client learning rate | Exploration param |
|---|---|---|---|
| Init (supervised) | 0.05 | 0.05 | · |
| Softmax | 0.05 | 1 | $\beta = 0.1$ |
| FALCON | 0.05 | 0.05 | $\mu = 10, \gamma = 1000$ |
| Greedy | 0.05 | 0.1 | · |
| $\epsilon$-Greedy | 0.05 | 0.05 | $\epsilon = 0.05$ |

Table 8: Hyperparameter settings for the SO dataset and the init-shift scenario with deploy_freq = 40 (Fig. 2d)

| Dataset | Clip Norm | Noise Multiplier | Server learning rate | Client learning rate |
|---------|-----------|------------------|----------------------|----------------------|
| EMNIST | 0.1 | 0 | 0.002 | 0.2 |
| | | 0.01 | 0.002 | 0.1 |
| | | 0.1 | 0.002 | 0.2 |
| SO | 0.8 | 0 | 0.02 | 0.5 |
| | | 0.3 | 0.01 | 2 |
| | | 0.7 | 0.01 | 2 |

Table 9: Hyperparameter settings for the DP experiments using `Softmax` $\beta = 0.05$ in the `scratch` scenario (Fig. 4).

