# OpenReview forum: "An Empirical Evaluation of Federated Contextual Bandit Algorithms"
_TMLR — Withdrawn by Authors_

### Review · Reviewer_R1qQ · 2023-05-22

**Summary Of Contributions:**

The paper presents an emirical evaluation of bandit algorithms in the federated learning setting. 4 algorithms (softmax, greedy, FALCON, and epsilon-greedy) are compared on two domains (EMNST and SO, that are supervised learning domains turned into contextual bandit domains). The bandit algorithms are wrapped with the FedAvg algorithm to deploy them in the federated setting. Further, differential privacy extensions are also presented.

**Audience:**

Yes

**Broader Impact Concerns:**

-

**Claims And Evidence:**

No

**Requested Changes:**

Overall,

A. I think the paper needs to be revised to be more precise and make the contribution upfront.
Currently, the paper is too verbose for the contribution. Given that the main contribution is experimental benchmarking, the first new takeaway point only starts on page 12.

B. Further, given that the topic is about federated learning, it would be beneficial if the insights are tailored for the federated learning setting. In the current form, most of the insights seem generic.

B.1 There are several federated learning algorithms out there, it might be worth trying to see the impact of using different ones and understanding their pros and cons in the bandit setting.

B.2 Similarly, for differential privacy, there are several methods out there and it is not clear which one is more appropriate for the bandit setting.

B.3 An important topic in federated learning is further to minimize the communication cost, this can be done by either (a) minimizing the frequency at which the information is shared (e.g., delayed communication), or (b) minimizing the size of the information shared during every communication (e.g., gradient compression). This aspect has largely been ignored in this work.

B.4 Effect of asynchronous updates to the server: Currently, the central server waits until all the clients respond. This might not be a scalable approach. It is worth understanding what is the impact of asynchronous updates for the bandit setting.

B.5 An important research direction when working in federated learning is making the system robust to byzantine actors. It is worth exploring the robustness of the CB methods to these modes of attack, and adapting solutions from the federated learning literature to see if they help in the CB setting.

B.6 Currently, it is not clear how non i.i.d. is the data across different clients. Varying this and seeing the impact of the above settings is also important  for the federated learning setting.

C. A common complaint of using supervised learning domains to do bandit experiments is the lack of diversity in reward variances across different arms. Having datasets that lots of actions, and with varying distributions of reward would make the experiments stronger.

D. Effect of the optimizer, I am curious if the authors also explored using ADAM or other adaptive optimizers at the client side? Maybe that increases the communication cost as one might have to relay statistics beyond just the gradient Given that ADAM has a tangible impact when used on the server side, it might be natural to experiment on using it on the client side as well.

**Strengths And Weaknesses:**

Strength:

1. Considers and important and understudied topic of contextual bandit in the federated learning setting.

Weakness:

1. Insights are more general and less focused on nuances of federated learning, given that federated learning is the topic of the paper.
2. Few and simple domains, especially for a paper focused on empirical evaluations.

---

### Review · Reviewer_qsRr · 2023-05-28

**Summary Of Contributions:**

This paper studies the federated contextual bandit problem. The authors design several variants of contextual bandit algorithms from the centralized setting to adapt to the federated setting. The authors provide empirical evaluations on the public datasets in federated learning. The simulation setups model real-world federated learning scenarios, including the misalignments between an initial pre-trained model and the subsequent user interactions due to data non-stationarity, and the heterogeneity across clients. The experimental results reveal the effectiveness of the softmax heuristic in balancing the exploration and exploitation.

**Audience:**

Yes

**Broader Impact Concerns:**

I believe that this paper does not have negative ethical impacts.

**Claims And Evidence:**

Yes

**Requested Changes:**

1.	The authors should give more motivating scenarios where we need to incorporate contextual bandits in the federated learning formulation.
2.	It would improve this paper if the authors give more comparisons on formulations, algorithms and results with existing federated/distributed contextual bandit works.
3.	I suggest the authors to present their theoretical results (Section 6) by formal theorems or lemmas, and add more discussion on the implications of the results. It would be better if the authors include more comparisons with the regret bounds of existing federated/distributed bandit algorithms.
4.	The authors should elaborate more about the unique challenges and novelty of their federated contextual bandit formulation and algorithms.



**Strengths And Weaknesses:**

Strengths:
1.	The authors conduct empirical evaluations on two public datasets with a wide range of simulation setups. The empirical results demonstrate the effectiveness of the softmax approach and the importance of variance control.
2.	This paper is well organized and clearly written.

Weaknesses:
1.	The motivation of the combination between federated learning and contextual bandits is not clear. The authors should elaborate more about the real-world distributed/federated applications where we need to interact with the environment to obtain samples. What is the unique challenges of federated contextual bandits compared to existing federated learning and distributed bandit formulations?
2.	While the authors conduct empirical evaluations on a wide range of setups, the theoretical analysis part looks weak. In Section 6 (theoretical analysis), the authors do not provide any formal theorems or lemmas, which make the theoretical contributions of this paper unclear. While the authors demonstrate the effectiveness of the softmax approach empirically, the theoretical result for the softmax approach is missing. The authors should provide more discussion on the regret bounds and more comparisons with the results of existing federated learning/distributed bandit algorithms.
3.	The technical novelty of algorithm design and theoretical analysis for federated contextual bandits is unclear. Several algorithm components, e.g., \epsilon-greedy, softmax and importance weighting, are not new. What is the main novelty of this paper compared to prior federated learning/distributed bandit works?

---

### Review · Reviewer_j8ZA · 2023-05-31

**Summary Of Contributions:**

This paper studies the federated contextual bandits problem, where the federated setting is centralized and cross-device setting. As the feedback is only observed for the selected action in the contextual bandit problem, the paper uses a heuristic approach (i.e., softmax heuristic) for dealing with the exploration-exploitation tradeoff. To evaluate the performance of the proposed softmax heuristic, the authors have considered different settings for experiments on two real datasets: EMNIST and StackOverflow.

**Audience:**

Yes

**Broader Impact Concerns:**

I do not find any ethical concerns.

**Claims And Evidence:**

Yes

**Requested Changes:**

1. Compare comparison with existing contextual bandits algorithms (especially UCB- or TS-based CB algorithms) and add explanations why they are omitted from the experiments.

2. "We limit the inference to only happen at the clients selected for training at this round, since the inference data generated at the other clients does not matter for model updates.": explain why data (generated for other clients) cannot be used when these clients are selected in the future. Also, explain why the updated model is used in Line 6 (FL update) but not for inference (Line 5) in Algorithm 4.

3. Does $\rho$ used in Eq. (6) the same for all actions and contexts? If not, the authors should use the notation that may avoid this confusion, e.g., $\rho_{x,a}$

4. The authors can also add experiments showing how minibatch size influences the regret

**Strengths And Weaknesses:**

#### **Strengths of paper:**
1. This paper considers the federated contextual bandits problem in a cross-device and centralized federated setting. This problem has many practical applications. e.g., recommendation algorithm for users without sharing their data.

2. To deal with the exploration and exploitation tradeoff in a bandit setting, the paper proposed a softmax heuristic, which is a simple method.

3. The authors have considered different practical settings for the performance evaluation of their proposed method. These settings can be used as a performance evaluation setup for future federated contextual bandits algorithms.

#### **Weaknesses of paper:**
1. There is rich literature on contextual bandits, where different algorithms are proposed to deal with the exploration and exploitation tradeoff, e.g., upper confidence bound (UCB) and Thompson Sampling (TS). There needs to be a discussion of these works and why they are not considered in the experiments.

2. The statement in Line 3 of the last paragraph on Page 2, "... impracticality of updating the CB model after each example... " is unclear to me as many existing contextual bandits algorithms update contextual bandits (CB) model, e.g., Lin-UCB, Neural-UCB, etc.

3. It is unclear how the 'softmax heuristic' will work if the action space is continuous (how does line 5 of Algorithm 2 work?). Further, it is also unclear how the choice of $\beta$ influences the regret.

---

### Note · Authors · 2023-06-07

**Comment:**

We thank the reviewers for their valuable feedback.

While we intend to address the concerns on writing and presentation, it looks like many additional experiments are requested, which can not be easily done within a reasonable timeline.

We also feel some of the comments requesting comparison to previous methods, or combine with various methods are vague, and not quite actionable without providing a few concrete references. The requests have a wide coverage on optimization, DP, robustness and so on, and many of the studies can probably become a new paper. This paper is intended to be an initial exploration for bandits algorithms in relatively practical settings of FL instead of a textbook.

**Withdrawal Confirmation:**

I have read and agree with the venue's withdrawal policy on behalf of myself and my co-authors.